# Response to Immune Checkpoint Inhibitors Is Affected by Deregulations in the Antigen Presentation Machinery: A Systematic Review and Meta-Analysis

**DOI:** 10.3390/jcm12010329

**Published:** 2022-12-31

**Authors:** Maria Rasmussen, Jon Ambæk Durhuus, Mef Nilbert, Ove Andersen, Christina Therkildsen

**Affiliations:** 1Department of Clinical Research, Copenhagen University Hospital—Amager and Hvidovre, Hvidovre, 2650 Copenhagen, Denmark; 2Center for Healthy Aging, Department of Cellular and Molecular Medicine, University of Copenhagen, 2200 Copenhagen, Denmark; 3Institute of Clinical Sciences, Division of Oncology and Pathology, Lund University, SE-22185 Lund, Sweden; 4The Danish HNPCC Register, Gastro Unit, Copenhagen University Hospital—Amager and Hvidovre, Hvidovre, 2650 Copenhagen, Denmark

**Keywords:** immunotherapy, immunoediting, antigen presenting machinery, biomarkers, melanoma, non-small cell lung cancer

## Abstract

Immune checkpoint inhibitors (ICI) targeting programmed death 1 (PD-1), its ligand (PD-L1), or cytotoxic T-lymphocyte antigen 4 (CTLA-4) have shown promising results against multiple cancers, where they reactivate exhausted T cells primed to eliminate tumor cells. ICI therapies have been particularly successful in hypermutated cancers infiltrated with lymphocytes. However, resistance may appear in tumors evading the immune system through alternative mechanisms than the PD-1/PD-L1 or CTLA-4 pathways. A systematic pan-cancer literature search was conducted to examine the association between alternative immune evasion mechanisms via the antigen presentation machinery (APM) and resistance towards ICI treatments targeting PD-1 (pembrolizumab and nivolumab), PD-L1 (durvalumab, avelumab, and atezolizumab), and CTLA-4 (ipilimumab). The APM proteins included the human leucocyte antigen (HLA) class I, its subunit beta-2 microglobulin (B2M), the transporter associated with antigen processing (TAP) 1, TAP2, and the NOD-like receptor family CARD domain containing 5 (NLRC5). In total, 18 cohort studies (including 21 original study cohorts) containing 966 eligible patients and 9 case studies including 12 patients were reviewed. Defects in the APM significantly predicted poor clinical benefit with an odds ratio (OR) of 0.39 (95% CI 0.24–0.63, *p* < 0.001). The effect was non-significant, when considering complete and partial responses only (OR = 0.52, 95% CI 0.18–1.47, *p* = 0.216). In summary, the APM contains important targets for tumorigenic alterations which may explain insensitivity towards ICI therapy.

## 1. Introduction

The human body hosts a natural immune defense against cancer cells, in which the cytotoxic T cells recognize and kill cancer cells presenting mutations as neo-epitopes on their cell surface via the human leukocyte antigen (HLA) class I receptors. Hence, tumor-induced immune evasion through exhaustion or inactivation of these T cells is an essential step during tumorigenesis [1]. One evasion mechanism utilized by the cancer cells is up-regulation of programmed death 1 ligand (PD-L1), which inhibit the cytotoxic T cell activity (i.e., exhaustion) [1,2] (Figure 1).

Drugs targeting programmed death 1 (PD-1) receptor or its ligand (PD-L1) abrogate this immune evasion mechanism and have shown promising responses in multiple cancer types [3,4,5,6]. Furthermore, drugs may activate the T cells through secondary stimulation by blocking the cytotoxic T-lymphocyte-associated protein 4 (CTLA-4) [2,7]. Currently, treatment with such immune checkpoint inhibitors (ICI) is part of standard therapies in multiple malignancies, including malignant melanoma, lung cancer, renal cell carcinoma, bladder cancer, head and neck cancers, and colorectal cancer [8].

Although good responses in case reports and early sub-cohorts from ongoing clinical trials have been observed [4,6,9,10,11], increasing numbers of treated cancer patients and extended follow-up have led to decreasing clinical benefit [12,13,14]. Despite the focus on biomarkers associated with response, the majority of patients do not respond to ICI [12,13,15,16,17] which can be due to innate resistance (primary) or acquired resistance (secondary) [18]. Focus should therefore be added to biomarkers of non-response since it only takes one deregulated protein to mediate (clonal) resistance despite having one or multiple biomarkers associated with response. Hence, focus on the biomarkers predicting resistance is required to improve the selection of cancer patients to ICI therapy.

Such possible biomarkers may be found within the antigen processing and presentation pathway (also named the antigen presentation machinery, APM), which loads the neo-epitopes into the classical HLA class I receptor and transports them to the tumor cell surface, where they can be recognized by cytotoxic T cells (Figure 1). Besides the HLA class I genes, important APM genes that may be mutated during tumorigenesis include the transporter associated with antigen processing (TAP) 1 and TAP2, which loads the foreign peptide fragments in the HLA class I receptors; the HLA class I transcription factor NOD-like receptor family CARD domain containing 5 (NLRC5); and the essential HLA class I subunit, beta-2 microglobulin (B2M) [19]. Somatic mutations and/or loss-of-heterozygosity (LOH) in the APM genes may result in lack of antigen presentation and thereby lack of cytotoxic T cell-mediated tumor elimination (Figure 1). 

We have systematically reviewed the clinical benefit among patients with tumorigenic APM defects treated with the anti-PD-1, anti-PD-L1, and/or anti-CTLA-4 antibodies pembrolizumab, nivolumab, durvalumab, avelumab, atezolizumab, and ipilimumab and found that the deregulations within the APM may explain lack of response towards ICI therapy.

## 2. Materials and Methods

### 2.1. Systematic Literature Search

A systematic literature search was performed to identify cancer patients treated with anti-PD-1 (pembrolizumab, nivolumab), anti-PD-L1 (avelumab, durvalumab, atezolizumab), or anti-CTL-4 (ipilimumab) drugs where molecular investigations including the following APM biomarkers: classical HLA class I (A, B, and C), B2M, TAP1, TAP2, and/or NLRC5, had been conducted and put in relation to response to treatment. Only studies of non-hematologic cancers were eligible for this study.

The search string consisted of four categories: a cancer category [“Neoplasms” OR “Cancer” OR “Tumor”], an immunotherapy category [“Ipilimumab” OR “Nivolumab” OR “Pembrolizumab” OR “Avelumab” OR “Atezolizumab” OR “Durvalumab”], an antigen-processing and presentation biomarker category [“B2M” OR “HLA-A” OR “HLA-B” OR “HLA-C” OR “TAP1” OR “TAP2” OR “NLRC5”], and a clinical outcome category [“Survival” OR “Objective Response” OR “Disease progression” OR “Disease regression” OR “Tumor progression” OR “Tumor regression” OR “Drug resistance”]. The four categories were combined with a Boolean “AND” and all search terms were included as MeSH terms or Supplementary Concept when possible, and Text Word (combined with “OR”) to identify yet unindexed articles. All studies had to be on Homo sapiens and written in English. The final search was conducted on 8 December 2021, in PubMed and can be shared upon request from the authors. The search results were imported into Rayyan QCRI application (https://www.rayyan.ai/, acceded on 8 December 2022) [20]. No protocol was made for this systematic review.

### 2.2. Data Extraction

The search results were all screened on title and abstract level, followed by full text reading of selected articles, according to the inclusion and exclusion criteria. This was done by three reviewers (M.R., J.A.D., and C.T.) with at least two reviewers for each study. In case of discrepancy, the third reviewer was consulted to reach agreement. The following data were extracted: country of study, study design, year published, study population (number of patients treated with ICI and number of treated patients with relevant biomarker analyses and treatment response), cancer type, ICI treatment, APM biomarkers investigated, methods used for biomarker investigations, and clinical outcome (objective response or equal and overall and progression-free survival) correlated to the relevant biomarker(s). If information were available regarding age and sex for the eligible patient cohorts, these data were also extracted. Data from one cohort study was included as a case report as the biomarkers of interest had only been studied for one individual [21].

Quality assessment was performed with high risk of bias defined as studies with APM biomarker analyses performed in cohorts and cases ascertained for progressive disease (PD) during ICI therapy. Studies were requested to be original articles, but in six studies the authors had re-analyzed genetic arrays from previously published cohort studies [22,23,24,25,26,27], where three were not identified in the original search [28,29,30] (Figure 2).

### 2.3. Definitions

Eligible patients should have been treated with ICI with availability of tumor response data that could be correlated with data from APM biomarker analyses of the corresponding tumor. Normal APM/APM positive was defined as non-mutated APM genes (for DNA analyses) and/or normal or high expression (for RNA and protein analyses). Defect APM/APM negative was defined as mutated APM genes (for DNA analyses) and/or low or loss of expression (for RNA/protein analyses). When possible, the distinction between APM positive and negative samples was used as described in the included studies. For five studies, these data were extracted by the reviewers from heatmaps and figures presented in the studies [23,24,26,27,31]. When evaluating heatmaps with several relevant genes, APM was scored as negative if one of the genes showed negative expression. Expressions at the reference level (white or black) were scored as normal [23,26,27]. When evaluating DNA mutation profiles with several relevant genes, APM was scored negative when at least one of the genes were mutated [31]. When DNA profiles included loss of heterozygosity analyses, APM was scored negative when at least one event occurred in one gene [24].

### 2.4. Clinical Outcome

The primary clinical outcome in this review was objective response rate (ORR), as the topicality of this research area entailed relatively short follow-up and few survival data. Progression-free survival (PFS) and overall survival (OS) was, however, also extracted whenever available for the two subgroups: APM positive and APM negative. Objective response rates were defined as the percentage of patients with partial response (PR) and complete response (CR). Clinical benefit was calculated as the percentage of patients with CR, PR, and stable disease (SD). PFS was measured in months from first dose of ICI to tumor progression or end of follow-up, whatever came first. OS was measured in months from the first dose of ICI to death or end of follow-up. Half of the cohort studies scored the tumor response at the end of treatment with ICI (final response), while the other half used best overall response during ICI therapy as the final response, which may have skewed the results to identify more responders with APM defective tumors. When both best overall response and final response were possible to extract, the final response was used, except for the studies where patients were selected based on progression after initial response [32,33,34] or including data only for patients with progression [35]. But these were not included in the meta-analyses. Whenever possible, original data were used for the analyses. In studies missing ORR but with detailed accessible data, ORRs were calculated by the reviewers using these, which is mentioned in Table 1.

Responses were measured using the response evaluation criteria in solid tumors (RECIST v1.1) guidelines, immune-related response criteria (irRC), immune-modified RECIST (iRECIST), or pathologic complete response (Appendix A). Only responses in the tumors tested for the relevant APM biomarkers were noted, e.g., recurrent tumor, metastases, or primary tumor. Hence, responses in new primary tumors or metastases occurring in other organs during treatment, but not tested for the relevant biomarkers, were not considered. In one case, the biomarker-tested tumor was a basal cell carcinoma on the nose developing during treatment of non-small cell lung cancer (NSCLC), which was not analyzed for the biomarkers [44]. As response data were available for the basal cell carcinoma as well, we have reported these tumor data for this individual. For one case, both a primary malignant melanoma and an associated lung metastasis were investigated for the relevant biomarkers [45]. This was reported as two separate tumors in Table 2 but shown as a combined patient track in the swimmer plot.

### 2.5. Redundant Data Sets

Several of the cohort studies included had revisited and reanalyzed previously published and publicly available gene expression, mutation, or loss of heterozygosity data sets [24,25,26,27]. Both Thompson et al., Shim et al., and Sade-Feldman et al. included their own cohort in addition to re-visiting publicly available datasets previously published to perform new bioinformatic analyses, while Yoshihama et al., only re-analyzed data. When redundant data were published in more than one study, we included the study with the largest data set and excluded the others. The Van Allen study cohort [36] was used in three studies [24,25,27]. As the largest data set was obtained through the study by Sade-Feldman et al., this was included. In addition, the original study from Hugo and colleagues [28] did not include relevant data but the genetic arrays of the study cohort were re-analyzed in two studies [24,26], of which Sade-Feldman et al. had performed bioinformatic analyses on the largest study cohort, why these results were included instead. As Sade-Feldman and colleagues also presented data from their own cohort, data from three independent study cohorts were described in here. In addition to data from Hugo et al., Thompson et al. also included a study cohort from Huang et al. [29] and their own cohort and the two latter study cohorts were included. Yoshihama et al. included data previously published by Gide et al. [30] in addition to Hugo et al. As relevant data could not be found in Gide et al. we used the data from Yoshihama and colleagues. In addition, Balatoni et al. [52] was used to extract cohort information for the cohort analyzed in Ladányi et al. [22].

### 2.6. Statistical Analyses

Data on response from the cohort studies were imported into R Statistics (version 3.6.1, R Core Team, Vienna, Austria) using RStudio (RStudio Inc., Boston, MA, USA), from where all statistical analyses were made. Studies, where the cohorts were ascertained on PD during ICI treatment, were not used for meta-analyses. The remaining studies were pooled according to clinical outcome (clinical benefit or ORR) and data were stratified by APM status (positive or negative). Meta-analyses were conducted using the “metafor” package (version 3.4-0) [53]. The pooled odds ratio (OR) estimates were investigated for significance using the random effects model, due to differences in the clinical setup, and a restricted maximum likelihood estimator. Heterogeneity among studies was investigated using the I^2^-statistics with *p* < 0.05 indicating that there was no statical evidence for heterogeneity. Sensitivity analyses were performed using the random effects model and restricted maximum likelihood estimator, except for Busch et al., Huang et al., Sade-Feldman et al., and Chen et al. regarding clinical benefit where a maximum likelihood estimator was used instead [24,29,41,42]. Meta-analyses data were visualized using forest plots, while case report data were visualized using swimmer plots. Upon visualization of cases in the swimmer plot, consecutive or combined ICI therapies were pooled as one treatment. Non-ICI treatments prior and/or following ICI treatment were not included in the swimmer plot.

## 3. Results

### 3.1. The Literature Search

The systematic literature search identified 184 studies, which were all reviewed on the title and abstract level according to PRISMA guidelines. Of these, 43 were reviewed on full text level and 27 studies were included based on available data and correct study design (Figure 2). The studies included 8 case reports and 1 case series including 12 individual patients [21,44,45,46,47,48,49,50,51] as well as 18 cohort studies investigating 21 original cohorts [22,23,24,25,26,27,31,32,33,34,35,37,38,39,40,41,42,43] with a total number of 1498 patients. The studies were published from 2016 to 2021 (Appendix A).

### 3.2. Cohort Presentation

Twenty-one original study cohorts were analyzed in the 18 cohort articles, covering 1498 patients treated with ICI with data on response and APM biomarker status available for 966 patients. Studies on malignant melanoma (*n* = 11) or NSCLC (*n* = 6) patients were most frequent (Table 1).

ORR or clinical benefit to ICI treatment could not be calculated for two of the cohorts [37,43] but were reported as summarized results in Table 1. ORR and clinical benefit were extracted for 11 study cohorts [22,23,25,32,33,34,35,38,39,40,42] and 18 study cohorts [22,23,24,25,26,27,31,32,33,34,35,39,40,41,42], respectively. In total, 84 out of 226 (37.2%, range 0–100%) patients with APM negative tumors showed clinical benefit to ICI therapy compared to 248 out of 439 (56.5%, range 25–100%) among the APM positive. Likewise, the ORR was 28.2% (range 0–100%) for the APM negative patients and 35.7% (range 11–50%) for APM positive patients. 

#### Meta-Analyses

To estimate the response rate among unselected and unbiased cohorts, we removed data sets that had focused their APM biomarker analyses on tumors progressing during ICI treatment [32,33,34,35]. In the remaining studies, 76 out of 218 (34.9%, range 0–100%) patients with APM negative tumors showed clinical benefit to ICI therapy compared to 235 out of 426 (55.2%, range 25–100%) among the APM-positive.

Meta-analyses showed a significant clinical benefit among patients with APM positive tumors (OR = 2.59, 95% CI 1.59–4.23, *p* < 0.001, I^2^ = 26.70%, *p* = 0.183) (Figure 3). When only considering CR and PR (ORR), the OR lowered to a non-significant 1.92 (95% CI 0.68–5.42, *p* = 0.216, I^2^ = 55.42%, *p* = 0.058) (Figure 4). Sensitivity analyses did not affect the results or significance (with OR ranging from 2.39–3.28 and 1.37–2.59 for clinical benefit and ORR, respectively). Although detailed response data could not be extracted from Sinn and colleagues, OR was presented in the study as 2.8–3.2 in relation to TAP1, HLA-A, and HLA-B RNA expression depending on the specific biomarker [43].

### 3.3. Case Presentations

We included all relevant case reports containing data on APM biomarkers and clinical outcome to increase data completeness. Twelve cases from nine case studies fulfilled the inclusion criteria and data on responses to ICI treatment as well as APM biomarker analyses were extracted (Figure 5, Table 2). Besides one case series that presented four cases with Merkel cell carcinomas (and metastases) [50], four patients had malignant melanomas (or associated metastases), two had lung cancers, one had colorectal cancer, and one presented with an extrahepatic cholangiocarcinoma. The last patient had a lung cancer, but during ICI treatment a basal cell carcinoma on the nose developed, and as this tumor was the only one being subjected to APM biomarker analyses, response data related to this tumor was used for this study. Nivolumab was used for treatment for 8 patients, either alone (*n* = 5) or in combination with ipilimumab (*n* = 2) or pembrolizumab (*n* = 1). Pembrolizumab was used as monotherapy in three patients and in combination with Talimogene laherparepvec (TVEC) in one patient, while the last patient was treated with avelumab. Beside one patient, who was initially treated with a vaccination-approach against neo-peptides [21], patients were treated with non-immune related anti-cancer drugs prior to ICI treatment or following disease progression during ICI therapy.

#### 3.3.1. Response to ICI Therapy

Clinical benefit was experienced for 4 out of 12 (33.3%) patients with a CR in one patient at 21 months of follow up (Figure 5, Table 2) [44,45,46,51]. The malignant melanoma-induced lung metastasis in this patient was, however, only treated with pembrolizumab for 2 months before switching to TVEC combined with pembrolizumab based on biomarker analyses and the clinical benefit may thus not be caused by ICI treatment alone. The remaining 8 patients deteriorated despite ICI therapy (two died during follow up). Median PFS in patients with clinical benefit was 16 months (range 12–21) compared to 9 months (range 1–30) for the progressing cancer patients.

#### 3.3.2. ICI Resistance Acquired via APM Regulations

When the clinical responses were evaluated according to the results from APM biomarker analyses, only one case presented with a tumor that had normal expression of the investigated APM biomarkers (i.e., HLA class I and B2M) [46]. Unlike most of the other cases, this tumor was investigated molecularly before initiation of ICI therapy. The tumor showed a PR at 2.7 months and SD at 13 months.

Two cases presented with mixed APM pattern switching from a positive ICI response in a tumor with normal B2M expression to lack of sensitivity towards the ICI therapy in tumors with B2M loss of function [21,45]. The first patient presented with a recurrent malignant melanoma and no genetic alterations in the B2M gene. Following 2 months of ICI therapy the patient experienced a CR [45]. Maintenance ICI therapy was continued for two years, but 6 months after end of treatment, there was a recurrence of the malignant melanoma and despite re-initiation of the previously successful combination of ICI treatment, the patient developed PD with metastasis in the lung and brain. The lung metastasis showed a frameshift loss-of-function mutation in B2M, which may explain the acquired resistance to ICI therapy. Hence, the patient was switched to TVEC combined with pembrolizumab off-label for two months before switching to temolozomide, which resulted in no sign of recurrence or active disease after 19 months. The CR of the lung metastasis in this case may therefore not be explained by the ICI therapy. Likewise, the second patient presented with a metastatic malignant melanoma that had developed during neo-epitope-vaccination therapy, in which the immune system of the patient was boosted by concentrated shots of foreign epitopes found in the patient’s primary tumor [21]. ICI therapy was initiated, but the patient died three months later. APM biomarker analyses showed lack of HLA class I protein expression and loss of both alleles of B2M. Interestingly, the pre-vaccinated tumor expressed B2M, while the post-vaccinated tumor did not, which could explain the initial response to immune-related therapy and the progression during ICI treatment.

The remaining nine cases showed defective APM (Figure 5) and in here only one patient (11.1%) responded to ICI therapy. This case developed a primary basal cell carcinoma on the nose during ICI treatment of a NSCLC [44]. APM biomarker analyses was only conducted for this tumor to elucidate reasons for this development, and lack of HLA and B2M protein expression was found on the tumor cells. The tumor was surgically removed, but reoccurred and was resected later, while the patient was continuing the ICI treatment. The cancer did not reoccur during the 31.5 months follow-up. Hence, this tumor showed a CR following the guidelines for our review, although these results and explanations for the clinical benefit should be carefully considered.

## 4. Discussion

As the response data to ICI treatment increase, it becomes clear that like for many other cancer therapies response-predictive biomarkers to focus the intension-to-treat population are required. Despite using such biomarkers like hypermutability, microsatellite instability, or PD-L1, more than half of the patients do not respond [12,13,15,16]. As the currently approved ICI drugs all aim to re-activate the T cells, selective pressure is added to tumor cells that evade the immune system. An essential mechanism is loss of functional expression of HLA class I receptors loaded with neo-epitopes from the tumor cell. This evasion mechanism can be obtained by deregulations in the entire APM with the most common genes affected being the classical HLA class I (A, B, and C), B2M, TAP1/2, and NLRC5 [19]. In here, we found that patients with deregulated APM had a worse response to ICI therapies (OR = 0.39 for clinical benefit and OR = 0.52 for tumor regression/ORR) compared to patients with normal APM.

Although many of the studies investigated in this review found APM deregulations to be linked to low response rates, the data are heterogenous with reports also on clinical benefit in APM negative (mutated or deregulated) tumors [25,38,42]. Reasons for this discrepancy could be the use of different molecular analysis platforms as different types of deregulations may affect the APM differently (from LOH and down-regulation to biallelic mutations and complete loss). Interestingly, Shim and colleagues only performed LOH analyses and did not find any significant association between APM LOH and response in their own cohort or the cohort previously published by Van Allen et al. [25]. In contrast, when both LOH and somatic mutations of the same APM genes were investigated by Sade-Feldman and colleagues, a significant association was identified in three independent cohorts (including Van Allen’s, Hugo’s and their own) [24]. Hence, to comprehend the true impact on HLA class I cell surface expression, DNA analyses (mutation or LOH) should be coupled with immunohistochemical protein analyses.

In addition to the biomarkers in this review, other genes can be relevant for deregulation of APM and resistance to ICI therapy. An APM gene expression signature of 18 different genes (including B2M, HLA A/B/C and TAP 1/2) has been combined with tumor mutation burden in an algorithm referred to as tumor immunogenicity score [17]. This algorithm was shown to improve prediction of response to ICI therapy in datasets from Van Allen et al., and Hugo et al. Other genes include the low molecular weight protein (LMP) 2, LMP7, and LMP10 which are immunoproteasomes, specifically up-regulated during intensified immune responses, that break down the altered proteins into neo-epitopes [54]. LMP2 and LMP7 were analyzed by Ugurel and colleagues and low expression of these were found in conjunction with low expression of HLA class I and B2M and could be up-regulated when treated with histone deacetylase inhibitors (HDACi) [50]. Likewise, it has been suggested that inactivating modifications in the JAK/STAT transducing pathway, involved with HLA transcription, may lead to resistance towards ICI therapy [24,34,54]. JAK1/2 mutations may, however, not be as frequent as HLA/B2M alterations, as a screening for JAK1/2 mutations in two unselected cohorts only found 1/23 (4%) malignant melanoma with a JAK2 mutation and 1/16 (6%) mismatch repair deficient metastatic colon cancers with a JAK1 mutation, although both were identified in non-responders [55]. To this end, it is important to emphasize that the impact of genetic alterations on the cell surface expression of neo-epitope presenting HLA class I receptors remains unknown, thus the linkage of deregulations in such genes should be interpreted with care.

In addition to the included PD-L1/PD-1 antibodies in this review, multiple antibodies are continuously developed, tested and approved, such as sintilimab (not approved yet), dostarlimab, and cemiplimab [6,41,56,57]. However, in theory these drugs should have similar effects on APM positive tumors and are assumed not to be effective in APM negative tumors. Notably, Luo et al. reported a case showing HLA class I LOH, experiencing 12 months of durable PR while treated with sintilimab combined with chemotherapy (etoposide-lobaplatin) and an antiangiogenic drug (anlotinib) [57]. Reasons for this response could be explained by the combined treatment with non-ICI drugs or the fact that only LOH analyses were applied.

Another way of re-activating the T cell-mediated tumor-attack could be through individualized vaccines in which in silico predicted and in vitro generated neo-epitopes (identified by mutational profiles of the tumors) are injected into the cancer patient. It is, however, still relevant to know the APM status, as loss of HLA class I receptors on tumor cells will cause loss of presentation of the neo-epitopes to the primed T cells. This is supported by a case reported by Sahin et al., where a patient with a APM positive malignant melanoma-associated metastasis was treated with such a mutanome vaccine [21]. The patient experienced PD and new tumor analyses showed loss of B2M protein expression emphasizing the selective pressure of APM negative tumor cells.

To improve response to ICI, drugs that can reverse immune evasion via down-regulation of APM protein could be considered. HLA class I can be epigenetically downregulation by acetylations, which can be reverted by HDACi [58]. Such combined therapies have been investigated by Ugurel et al., where 2 patients with low APM expressions were treated with HDACi and ICI. Unfortunately, only one tumor could be analyzed following treatment with HDACi and showed an enhanced HLA class I expression on the tumor cell surface. The patient died after four months [50].

Given the high frequency of HLA class I and B2M mutations in the unselected cohorts in this review (27.7%) and the hypothesis that the eventual second hit will lead to mutations in the APM genes, a natural killer (NK) cell-based therapy may be the obvious choice, since NK cells are destined to target foreign and human cells that do not express HLA class I receptor [59,60]. However, tumor-directed NK cell therapy to large solid tumors may be challenged by a low vascularity and low oxygen concentration, which in turn may cause a substantial decrease of NK cell activity [60].

The limited data on this topic motivated inclusion of studies disregarding tumor type and ICI therapy. The different combinations of ICI drugs and addition of other types of chemotherapies, angiogenic drugs, mutanome vaccines, and oncolytic viral gene therapy may all affect the clinical response in the patients, and it remains unresolved in these cases to which extend the ICI therapy failed or succeeded based on the APM status. Additionally, different APM biomarkers were chosen and analyzed using a variety of molecular approaches, and thus we pooled the results of all the studies to increase the power. Single loss of one allele may not have the same molecular effect on HLA class I expression as biallelic loss-of-function alterations. Hence, the overall heterogeneity of the included studies inhibits the establishment of specific recommendations, and these results should be interpreted with care. Although case reports may only be published when the results are interesting and thereby induce publication bias, we included both case reports and case series in addition to the cohort studies to ensure summary of all available cases.

It is possible that the search string used in this review may have introduced publication bias since studies investigating the APM biomarkers in focus and showing positive findings (such as resistance or tumor progression) are more likely to publish their results in abstract and title level. It is also possible that cancer patients with somatic alterations in the APM may respond to ICI therapy, and therefore simply not be genetically analyzed and thus miss inclusion in this review. Likewise, we may have missed studies on cancer patients with a good clinical response to ICI and normal APM activity and such cases are equally relevant to include to estimate the true predictive value of APM defects. The lack of such possible studies may have led to lower ORs in this review than the true estimate. On the other hand, half of the cohort studies used in the meta-analyses scored the tumor response at the end of treatment with ICI, while the other half used best overall response during ICI therapy, which may have skewed the results to identify more responders with APM defective tumors. To that end, further studies in large and unselected cohorts of ICI treated patients are warranted to fully explore the predictive value of APM defects under more homogeneous circumstances.

## 5. Conclusions

In summary, we found that patients with deregulated APM were more likely to not respond to ICI therapy and that biallelic APM mutations may be linked to resistance to treatment. Although more evidence is needed to pinpoint the essential APM players and their precise impact on HLA cell surface expression, HLA class I and B2M expressions should be investigated prior to initiating ICI therapy or when the patient experiences disease progression during ICI.

## Figures and Tables

**Figure 1 jcm-12-00329-f001:**
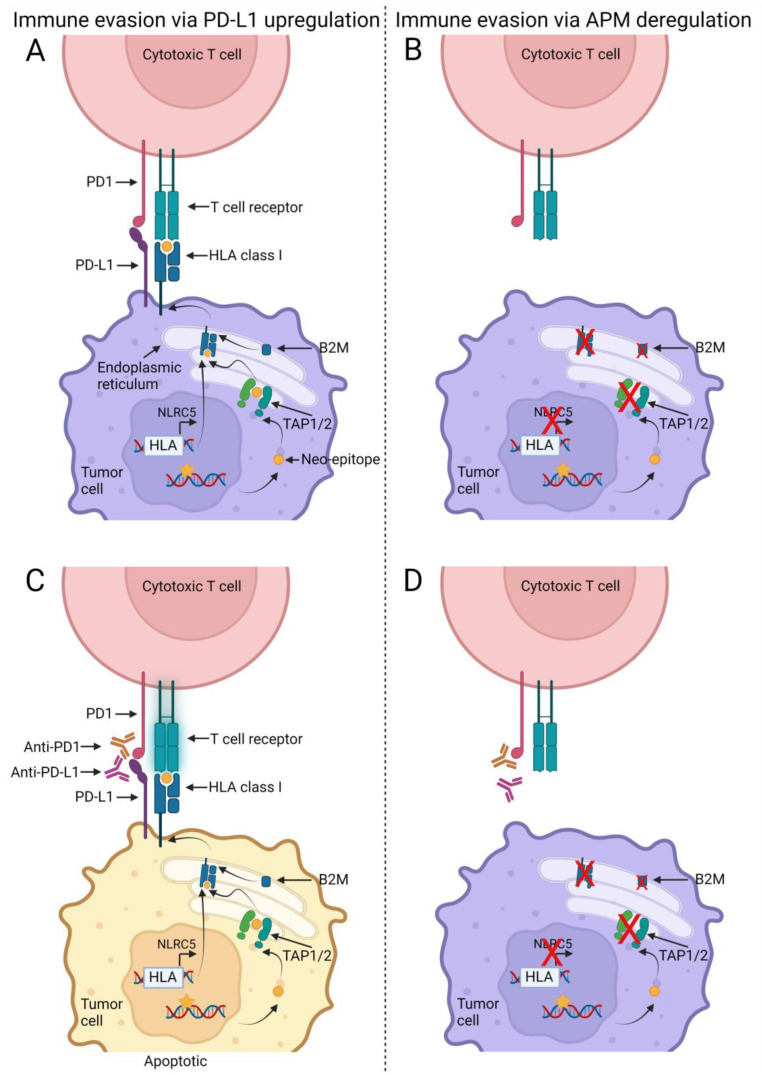
Model for non-responders and responders in relation to immune checkpoint inhibitors (ICI) and antigen presentation machinery (APM). DNA mutations (yellow star) generated by replicative errors during tumor cell division may result in altered protein structures. These proteins are degraded into smaller fragments (neo-epitopes) (yellow circle) that are imported into the endoplasmic reticulum and loaded onto the human leukocyte antigen (HLA) class I receptors by the transporter associated with antigen processing (TAP) 1 and TAP2. Prior to the loading the HLA class I molecule has been assembled with its essential subunit, beta-2 microglobulin (B2M). The assembled HLA class I receptor is transported to the cell surface where the T cell receptor on the cytotoxic T cells recognize the neo-epitope as foreign and in turn induce apoptosis in the tumor cell. The tumor cell can up-regulate the programmed death 1 ligand 1 (PD-L1) receptor and thereby inhibit the activity of the T cell (**A**). This interaction between PD-L1 and programmed death 1 (PD-1) can be inhibited by ICI therapy with anti-PD-L1 or anti-PD1 drugs and thereby re-activate the T cells (**B**). However, some tumors may evade the T cell mediated attack via deregulations of the APM proteins, such as B2M or HLA class I loss-of-function mutations (**C**). In theory, ICI therapy, reactivating the cytotoxic T cells, will not have any effect on the tumor cells as the T cells cannot identify the tumor cells (**D**). Created with BioRender.com, acceded on 18 November 2022.

**Figure 2 jcm-12-00329-f002:**
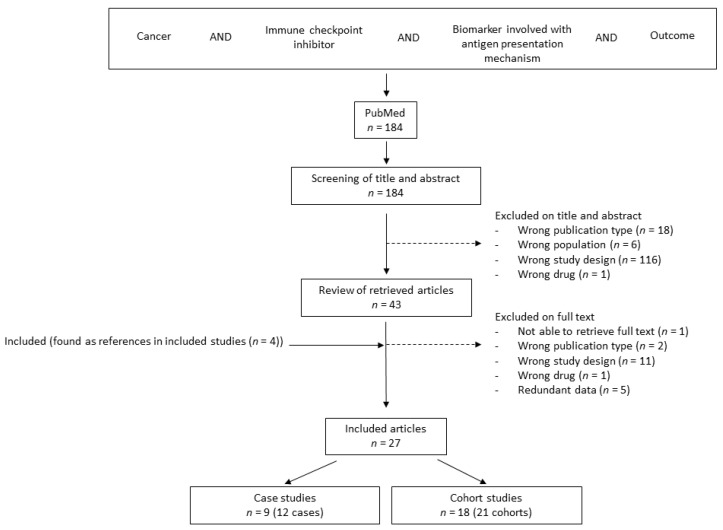
Flowchart depicting the inclusion, screening, and exclusion process of relevant studies. Wrong population included hematologic malignancies, while wrong study design included animal studies, cell line studies, no treatment, no relevant biomarker included, or no relation between biomarker data and response to ICI treatment.

**Figure 3 jcm-12-00329-f003:**
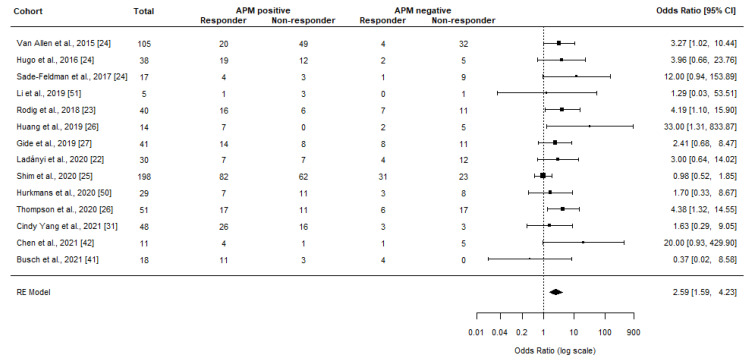
Meta-analyses investigating the association between normal APM (no mutations and/or normal RNA/protein expression of the relevant genes) and clinical benefit from ICI treatment. The black box represents the point estimate for the respective study, while the vertical line is the 95% CI. Gene Expression Omnibus (GEO) datasets from Hugo et al. [28] and Van Allen et al. [36] were re-analyzed in line with the objectives of this review in Sade-Feldman et al. [24]. Gide et al. [30] was reported in Yoshihama et al. [27], Huang et al. [29] was re-visited by Thompson et al. [26], while Balatoni et al. [52] was re-analyzed by Ladányi et al. [22].

**Figure 4 jcm-12-00329-f004:**
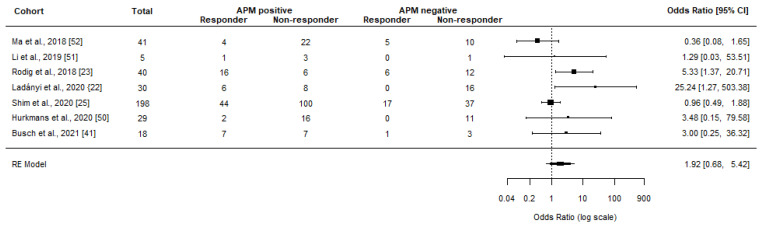
Meta-analyses investigating the association between normal APM (no mutations and/or normal RNA/protein expression of the relevant genes) and objective response rates from ICI treatment. The black box represents the point estimate for the respective study, while the vertical line is the 95% CI. Balatoni et al. [52] was re-analyzed by Ladányi et al. [22].

**Figure 5 jcm-12-00329-f005:**
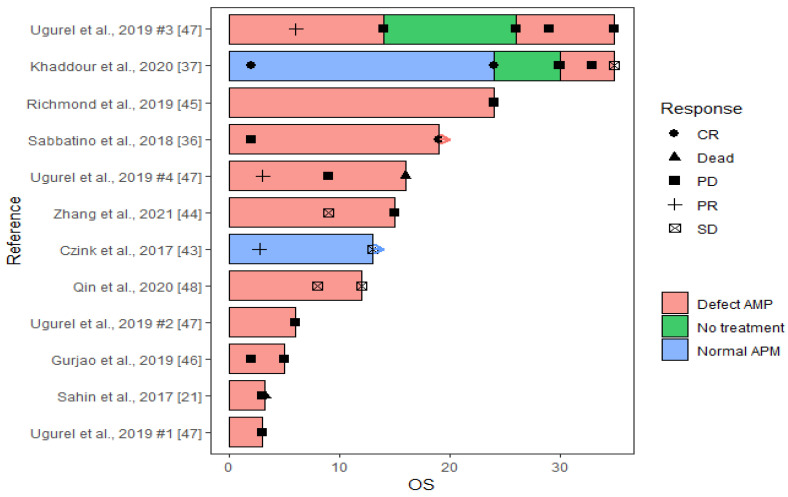
Swimmer plot of the 12 cases from the 9 case studies included. ICI therapies given as combinations or consecutive were merged as one treatment. Only PFS and OS time (months) are included in the plot, i.e., only time during ICI therapy. Hence, the initial response observed in the case from Sahin et al. [21], or the CR observed in Khaddour et al. [45] after removal of ICI drugs are not included. Arrows indicate that the ICI treatment is continued. A detailed description follows for complex cases. For Ugurel et al. [50], the four cases are referred to by #. For Ugurel et al., 2019 case #3, ICI treatment was terminated after tumor excision and the patient was followed with computed tomography until relapse after 12 months where ICI treatment was reinitiated. For Khaddour et al. [45], CR was observed after 2 months, and maintenance therapy was continued for 2 years. After 6 month the patient experienced a recurrence and treatment was reinitiated. After continued progressive disease, a B2M mutation was discovered, and ICI therapy was terminated after 5 months. The patient was then switched to chemotherapy. For Zhang et al. [47], ICI treatment was ended due to immune-related adverse events.

**Table 1 jcm-12-00329-t001:** Clinical response to ICI therapy in relation to defects in the APM (cohort studies).

Cohort [Reference with Relevant Data]	Patients (*n*) Treated/ Analyzed	Tumor Type	ICI Treatment	Response Definition	ORR	CB	PFS (Median Months)	OS (Median Months)	Biomarker	Summary	Comment
Van Allen et al., 2015 [36] ^a^	110/105	Metastatic melanoma	Ipi	CR, PR, and SD	NA	APM−: 11% APM+: 29%	NA	NA	Mutations and RNA expression of HLA-A, HLA-B, and HLA-C	B2M LOH was not significantly associated with survival	CR, PR, and SD or no clinical benefit with long-term survival
Hugo et al., 2016 [28] ^a^	38/38	Metastatic melanoma	Pembro or nivo	CR, PR, and SD	NA	APM−: 29% APM+: 61%	NA	NA	Mutations in B2M	B2M LOH was significantly associated with worse OS	
Zaretsky et al., 2016 [34] ^b^	78/4	Metastatic melanoma	Pembro	CR and PR	APM−: 100% APM+: 100%	APM−: 100% APM+: 100%	APM−: 19.5 APM+: 21.4	APM−: 31.5 APM+: 32.0	Mutations in B2M and NLRC5 and protein expression of HLA-A, HLA-B, and HLA-C	Patient 3 had a frame-shift deletion in B2M and loss of HLA class I protein expression. In supplementary data it could be seen that patient 2 had a missense mutation in NLRC5 in the relapse tumor. 2/4 with PR followed by PD, had a mutation in the APM	Responses and PFS recorded prior to PD
Seremet et al., 2016 [35]	39/4	Metastatic melanoma	Ipi or ipi + a dendritic cell vaccine	CR, PR, and SD ^c^	APM−: 0%	APM−: 0%	NA	APM−: 13	Protein expression of HLA class I and TAP1	Four patients (MEL15, MEL21, MEL22, and MEL26) with no clinical benefit showed HLA class I and/or TAP1 expression	HLA class I and TAP1 expression were unknown for the remaining samples
Johnson et al., 2016 [37]	30/28	Metastatic melanoma	Nivo or pembro or atezo	CR, PR, and SD ^d^	NA	NA	NA	NA	Protein expression of HLA-A	HLA-A expression level was not statistically associated with response to therapy	HLA-A positivity was not defined hampering detailed data extraction
Kakavand et al., 2017 [33] ^b^	44911	Metastatic melanoma	Pembro, nivo, or ipi + pembro	CR, PR, and SD ^d^	APM−: 100% APM+: 75%	APM−: 100% APM+: 100%	AMP−: 5.9 APM+: 14.9	NA	Protein expression of HLA-A	4/12 patients (33%) that progressed had a decrease of HLA-A protein expression	Responses and PFS recorded prior to PD
Ma et al., 2018 [38]	44/41	Recurrent or metastatic nasopharyngeal carcinoma	Nivo	CR, PR, and SD ^c^	APM−: 33% APM+: 15%	NA	APM−: 4.8 APM+: 1.8	APM−: NR APM+: 10.9	Protein expression of HLA-A and HLA-B	HLA-A and HLA-B expression did not predict response to nivo but there was a statistical association between loss of expression of HLA-A and/or HLA-B and PFS	Median 1-year PFS and OS
Sade-Feldman et al., 2017 [24]	17/17	Metastatic melanoma	Anti-CTLA-4, anti-PD1, or anti-PD-L1	Regression	NA	APM−: 10% APM+: 57%	NA	NA	Mutations in TAP1, TAP2, B2M, HLA-A, HLA-B, HLA-C, and protein expression of B2M and HLA-A, HLA-B, and HLA-C	Mutations in TAP1/2 were found in both non-responders and responders. 5 out of 17 patients (29%) exhibited B2M defects of which 3/5 initially responded and then progressed (Pat208 with LOH and frameshift mutation (and loss of protein expression, also of HLA class I); Pat33 with frameshift mutations; Pat99 with LOH and loss of HLA protein expression). 2/7 non-responders had B2M LOH (Pat25 (also with loss of HLA protein expression) and Pat115). Loss of both copies was only observed for non-responders. No B2M alterations were detected in responders within their cohort. There was no difference in HLA expression between responders and non-responders	CB was based on either somatic mutation or LOH or both in regard to progression/regression for all the relevant biomarkers and was scored by the authors
Li et al., 2019 [39]	60/5	Advanced NSCLC	Pembro	CR, PR, and SD	APM−: 0% APM+: 25%	APM−: 0% APM+: 25%	APM−: 7.0 APM+: 5.0	NA	Mutations in B2M	During the partial response for patient 8, two B2M mutations were identified. She had progressive disease after 30 weeks	
Rodig et al., 2018 [23]	280/181	Advanced malignant melanoma	Ipi, nivo + ipi	CR, PR, and SD ^c^	APM−: 33% APM+: 27%	APM−: 39% APM+: 27%	NA	NA	Protein and RNA expression of B2M, HLA-A, HLA-B, HLA-C, and TAP2	Reduced HLA class I expression was associated with primary resistance to ipi, but not to nivo. A gene set score derived from the top 25 differentially expressed genes (including TAP2) was significantly higher in tumor samples from patients without PD compared with those with PD at week 13 after single-agent nivolumab (CheckMate 064). For patients initially treated with ipi, low baseline tumor HLA class I expression (≤50%) was associated with inferior OS. No amount of tumor HLA class I expression identified a population with inferior OS when initially treated with nivo	Data from both CheckMate 064 and 069 were included. ORR and CB could, however, only be calculated for 40 patients from the CheckMate 064 study using data from TAP2 RNA expression scored by the reviewing authors
Huang et al., 2019 [29] ^e^	14/14	Advanced malignant melanoma	Pembro	No recurrence	NA	APM−: 29% APM+: 100%	NA	NA	RNA expression of B2M and TAP1 among others	Patients with no recurrence had a significantly higher APM score than those with recurrence with a median follow-up of 25 months. Disease-free survival was significantly longer in patients with upregulation in APM genes	APM+ was defined as an AMP Z score above 0
Giroux Leprieur et al., 2020 [32] ^f^	79/8	Advanced NSCLC	Nivo or pembro	CR, PR, and SD ^c^	APM−: 100% APM+: 100%	APM−: 100%, APM+: 100%	NA	NA	LOH of B2M, HLA-A, HLA-B	LOH of HLA-A and HLA-B in patient #1. LOH of B2M in patient #3	Responses recorded prior to PD
Ladányi et al., 2020 [22]	30/30	Metastatic melanoma	Ipi	CR, PR, and SD ^c,d^	APM−: 0%, APM+: 43%	APM−: 25% APM+: 50%	NA	NA	Protein expression of HLA-A, HLA-B, B2M	HLA class I antigen expression level in lymph node metastases, but not in cutaneous or subcutaneous metastases, was significantly correlated to clinical response and to patients’ OS. When evaluated in all metastases analyzed, it was not significantly associated with OS	APM− is defined as low expression of ≥2 APM biomarkers
Shim et al., 2020 [25]	198/198	Advanced NSCLC	Nivo, pembro, or anti-PD-L1	CR, PR, and SD ^d^	APM−: 32%, APM+: 31%	APM−: 57%, APM+: 57%	NA	NA	LOH of HLA class I	No association between HLA-LOH and response to the anti-PD1 /anti-PD-L1 agent. Reanalysis of the Van Allen cohort (110 melanoma patients) found the same	
Hurkmans et al., 2020 [40]	99/29	Advanced NSCLC	Nivo	CR, PR, and SD ^c,g^	APM−: 0% APM+: 11%	APM−: 27% APM+: 39%	NA	NA	Protein expression of HLA-A and HLA-B/C	HLA class I as an individual biomarker was not significantly associated with better OS or PFS. Patients with complete loss had impaired PFS compared to patients with partial loss or normal expression of HLA class I. No significant association was found for HLA class I and response groups	
Thompson et al., 2020 [26]	67/51	Metastatic NSCLC	Nivo, pembro, or atezo	CR, PR, and SD ^d^	NA	APM−: 26% APM+: 63%	APM−: 1.74 APM+: 18.1	APM−: 6.3 APM+: 19.7	RNA expression of B2M, TAP, and NLRC5	Higher expression of APM in the responder group compared with non-responders. APM score above the median value for the entire cohort was associated with significantly improved PFS and OS	For CB, downregulated expression was scored as APM− by the reviewing authors
Cindy Yang et al., 2021 [31]	106/48	30 advanced solid cancer types ^h^	Pembro	CR, PR, and SD ^c,i^	NA	APM−: 50% APM+: 62%	NA	NA	Mutations in B2M, TAP1, TAP2, and HLA-A	B2M LOH corresponded with resistance. No notable associations between the frequency of somatic LOH events in HLA class I genes and pembro therapeutic benefit	For CB, cases were scored by the reviewing authors as AMP- when at least one relevant gene was mutated
Chen et al., 2021 [41]	24/11	Metastatic NSCLC	Nivo or durva ^j^	CR, PR, and SD	NA	APM−: 17% APM+: 80%	NA	NA	RNA expression of HLA-A	Upregulation of HLA-A is associated with longer PFS and may be applied to predict the efficacy in patients with metastatic non-small cell lung cancer	
Busch et al., 2021 [42]	19/18	MSI metastatic GI cancers ^k^	Pembro or nivo + ipi	CR, PR, and SD ^c^	APM−: 25% APM+: 50%	APM−: 100% APM+: 79%	APM−: 19.5 APM+: 33.0	NA	Mutations in B2M	No significant differences in therapy best response and survival were observed between B2M-mutant tumors, all of which also had immunohistochemical loss of B2M, and B2M-wild type tumor patients	
Sinn et al., 2021 [43]	88/83	Early triple-negative breast cancer	Durva + chemotherapy	pCR	NA	NA	NA	NA	RNA expression of HLA-A, HLA-B, TAP1	High expression of eight genes (including TAP1, HLA-A, and HLA-B) were significantly associated with response	
Gide et al., 2019 [30] ^l^	63/41	Malignant melanoma	Nivo or pembro	CR, PR, and SD ^m,n^	NA	APM−: 42% APM+: 64%	NA	NA	RNA expression of HLA-A, HLA-B, HLA-C, B2M, TAP1, and NLRC5	The responder group exhibited higher expression of NLRC5 and HLA-B than the non-responder group, and B2M showed a similar trend although it was not statistically significant	For CB, downregulated expression was scored as APM− by the reviewing authors

Abbreviations: ICI, immune checkpoint inhibitor; ORR, objective response rate (complete response (CR) + partial response (PR)); CB, clinical benefit (CR + PR + stable disease (SD)); PFS, progression-free survival; OS, overall survival; pembro, pembrolizumab; nivo, nivolumab; durva, durvalumab; ipi, ipilimumab; atezo, atezolizumab; APM, antigen presenting machinery; APM−, APM negative; APM+, APM positive; LOH, loss-of-heterozygosity; NSCLC, non-small cell lung cancer; MSI, microsatellite instable; GI, gastrointestinal; pCR, pathologic complete response; OR, odds ratio; PD, progressive disease; NR, not reached; NA, not available. ^a^ Data re-analyzed and presented by Sade-Feldman et al., 2017. ^b^ Tumor progression after initial response. ^c^ Best overall response. ^d^ SD lasting at least 6 months. ^e^ Data re-analyzed and presented by Thompson et al., 2020. ^f^ Tumor progression after initial response to ICI lasting at least 6 months. ^g^ SD lasting at least 90 days. ^h^ Head and neck, triple-negative breast, high-grade serous carcinoma (HGSC), melanoma, and other mixed solid tumors. ^i^ SD longer than 18 weeks. ^j^ 2 of 11 unidentifiable cases treated with IBI308 (non-FDA approved anti-PD1 drug) could not be removed. ^k^ 15 colorectal, 2 gastric, and 1 cholangiocellular. ^l^ Data re-analyzed and presented by Yoshihama et al., 2021. ^m^ SD with overall survival greater than 1 year. ^n^ ORR and OR calculated based on summarized gene expression in Figure 5A in Yoshihama et al. [27].

**Table 2 jcm-12-00329-t002:** Clinical response to ICI therapy in relation to defects in the APM (case reports).

Case [Reference]	Tumor Investigated	ICI Treatment	Objective Response	PFS (Months)	OS (Months)	Biomarker Result	Comment
Czink et al., 2017 [46]	Recurrent extrahepatic cholangiocarcinoma	Pembro	SD	13	13	Normal expression of HLA class I and B2M	
Zhang et al., 2021 [47]	Lung adenocarcinoma	Nivo	PD	15	15	Homozygote HLA-B deletion	ICI treatment was ended due to adverse events
Richmond et al., 2019 [48]	Recurrent malignant melanoma	Nivo and pembro	PD	24	24	Loss of function B2M mutation	
Gurjao et al., 2019 [49]	Metastatic colorectal cancer	Pembro	PD	2	5	B2M frameshift mutation and LOH	
Ugurel et al., 2019 #1 [50]	Merkel cell carcinoma metastases	Avelu	PD	1 ^a^	1 ^a^	Loss of HLA class I protein expression	
Ugurel et al., 2019 #2 [50]	Merkel cell carcinoma	Nivo	PD	6	6	Low protein expression of HLA class I	
Ugurel et al., 2019 #3 [50]	Merkel cell carcinoma metastases	Nivo	PD	14	35	Low protein expression of HLA class I	
Ugurel et al., 2019 #4 [50]	Recurrent Merkel cell carcinoma metastasis	Pembro	PD	9	16	No HLA class I expression	
Sahin et al., 2017 [21]	Recurrent malignant melanoma metastases	Nivo	PD	3	3.2 ^b^	Homozygote B2M deletion	Part of a phase I study
Sabbatino et al., 2018 [44]	Basal cell carcinoma	Nivo	CR	19	19	Lack of HLA class I and B2M protein expression	Patient originally presented with a non-small cell lung cancer
Khaddour et al., 2020 [45]	Recurrent metastatic malignant melanoma	Combined nivo and ipi	CR	30	30	No B2M mutation	Patient had CR for 30 months before lung metastases developed at 30 months
Khaddour et al., 2020 [45]	Malignant melanoma-induced lung metastasis	Combined nivo and ipi followed by pembro (combined with TVEC and TMZ)	Good response	3	5	Loss of function B2M mutation	Pembrolizumab treatment was ended after 2 months due to verified B2M mutation
Qin et al., 2020 [51]	Metastatic large cell neuroendocrine carcinoma in the lung	Combined nivo and ipi	SD	12	12	B2M frameshift mutation in both primary and metastasis	

Abbreviations: ICI, immune checkpoint inhibitor; PFS, progression-free survival; OS, overall survival; pembro, pembrolizumab; SD, stable disease; nivo, nivolumab; PD, progresseive disease; LOH, loss of heterozygocity; avelu, avelumab; CR, complete response; ipi, ipilimumab; TVEC, Talimogene laherparepvec; TMZ, Temozolomide. ^a^ Estimate suggested by the authors based on treatment with avelumab every second week. ^b^ Estimate suggested by the authors based on rapid death after last PD. For Ugurel et al., 2019 [50], the four cases are referred to by #.

## Data Availability

The authors confirm that the data supporting the findings of this study are available within the article or its Appendix A. Detailed data extracted from the reviewing authors as well as the full search string are available upon reasonable request.

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
