# Peer review of "Response to Immune Checkpoint Inhibitors Is Affected by Deregulations in the Antigen Presentation Machinery: A Systematic Review and Meta-Analysis"

_jcm, 2022, doi:10.3390/jcm12010329_

Round 1

Reviewer 1 Report

Authors in this manuscript permformed a systemic reveiw on the connection between alternative mechanisms of immune evasion and resistance against immune checkpoint inhibitors. They reveiwed 18 cohort and 9 cases, and analyzed the data. The citaions used by authors are appropriate and up-to-date. The manuscript is well written and organized in general. There is a small error needed to be corrected: in the result section, the subtitle should be numbered: "3.1 The literature search", instead of "3.2".

Author Response

Dear reviewer,

Thank you very much for the positive and constructive feedback for the manuscript above.

Authors in this manuscript performed a systemic review on the connection between alternative mechanisms of immune evasion and resistance against immune checkpoint inhibitors. They reviewed 18 cohort and 9 cases, and analyzed the data. The citations used by authors are appropriate and up-to-date. The manuscript is well written and organized in general.

Point 1: There is a small error needed to be corrected: in the result section, the subtitle should be numbered: "3.1 The literature search", instead of "3.2".

Response 1: Thank you for noticing this. We have edited it.

Kind regards
Maria Rasmussen,
Corresponding author.

Reviewer 2 Report

Rasmussen et al presented a systematic review and meta-analysis in antigen presentation machinery status and response to immune checkpoint inhibitors. They concluded that patients with deregulated APM were more likely to not respond to ICI therapy and that biallelic APM mutations may be linked to resistance to treatment. Overall this study addressed an important issue in cancer immunotherapy, and some comments are listed as below: 

1, In Line 146, the authors define: Normal APM or APM positive was defined as non-mutated APM genes and/or normal-high expression of RNA and protein. However how to define “high/low expression of APM” is not described in the text. 

2, The critical function of APM in the clinical response to immune checkpoint inhibitors has been reported and studied previously, for example Wang et al Elife 2019 Nov 26;8:e49020. This background information should be mentioned in this related reports. 

3, Change "PD" to "progressive disease (PD)" in line 202, 

4, Change "3.2 The literature search" to "3.1 The literature search" in line 216. 

5, Table 1 is not well-presented, and should be adjusted to give a clear presentation. 

6, The presentation in Figure 5 appears chaotic and not concise, should also be adjusted.

Author Response

Dear reviewer,

Thank you very much for the positive and constructive feedback for the manuscript above.

Rasmussen et al presented a systematic review and meta-analysis in antigen presentation machinery status and response to immune checkpoint inhibitors. They concluded that patients with deregulated APM were more likely to not respond to ICI therapy and that biallelic APM mutations may be linked to resistance to treatment. Overall this study addressed an important issue in cancer immunotherapy, and some comments are listed as below: 

Point 1: In Line 146, the authors define: Normal APM or APM positive was defined as non-mutated APM genes and/or normal-high expression of RNA and protein. However how to define “high/low expression of APM” is not described in the text. 

Response 1: It is true that information on when a high or normal level is high or normal, e.g., using cut offs in concentrations etc. is missing from the text. However, as this review has covered a broad variety of studies – each performing different analyses and non-standardized scoring methods – it remains difficult to state exact cut offs for when a sample is considered APM positive. We have therefor used each study’s own definition of APM positivity, which was stated in the Method section. When data were available in the original studies, but not reported in the paper, the reviewing authors scored the AMP positivity/negativity by investigating heatmaps or DNA mutation profiles. This has now been described in detail in the Methods and added in the comment column of Table 1. For scoring using heatmaps, AMP positivity was defined as normal (white/black) or high (colored) expression, which has likewise been added to the Method section.

Point 2: The critical function of APM in the clinical response to immune checkpoint inhibitors has been reported and studied previously, for example Wang et al Elife 2019 Nov 26;8:e49020. This background information should be mentioned in this related reports. 

Response 2: Thank you very much for sharing this knowledge. The work by Wang et al is interesting and has been added to the review to all the relevant paragraphs.

Point 3: Change "PD" to "progressive disease (PD)" in line 202, 

Response 3: Thank you for noticing that this abbreviation had not been given in the text. The abbreviation was added in line134 instead, and “PD” in line 202 was kept. 

Point 4: Change "3.2 The literature search" to "3.1 The literature search" in line 216. 

Response 4: Thank you for noticing this. This has been changed.

Point 5: Table 1 is not well-presented, and should be adjusted to give a clear presentation. 

Response 5: We agree that the tables were difficult to assess in its past format and have now added the tables as figures to increase readability. However, we have communicated with the editors, and we will receive help to improve the tables in the right format.

Point 6: The presentation in Figure 5 appears chaotic and not concise, should also be adjusted.

Response 6: We are not sure that we understand completely what is meant by “chaotic”. Swimmer plots of case reports are always a bit chaotic and busy, and we have already tried to simplify the cases by pooling different ICI therapies into one single treatment regimen and not reporting responses or data from other treatment regimens, which explains why some of the cases end their lines end with a progressive disease that is untreated. We have added some details regarding the drawing of the swimmer plot in the “statistical analyses” in the Methods section and how complex cases were handled in the “clinical outcome” section under Methods. Furthermore, we have added information on complex cases in the legend to the swimmer plot to hopefully clarify the data. We agree that it does look odd that, e.g., the first case (in the top of the swimmer plot) from Ugurel #3 progress from the disease but is left without treatment for 12 months. Indeed, this case experienced a progressive disease, which was surgically removed and the patient was left without treatment of any kind until progression after 12 months, from where the ICI therapy was re-initiated. This and other cases have been explained in detail in the figure legend. Additionally, we identified a discordance between Table 2 and the swimmer plot for the case presented by Zhang et al, and Table 2 has now been corrected as the overall survival was only 15 months and progressive disease was recorded already at that time point.

Kind regards
Maria Rasmussen,
Corresponding author.

Reviewer 3 Report

The review article entitled "Response to immune checkpoint inhibitors is affected by deregulations in the antigen presentation machinery: a systematic review and meta-analysis" is very interesting. However, there are minor concerns regarding the Tables 1 and 2.  

Table 1. Clinical response to ICI therapy in relation to defects in the antigen presentation machinery (cohort studies).

Table 2. Clinical response to ICI therapy in relation to defects in the antigen presentation machinery (case reports).

These tables are not properly inserted in the text. Authors need to correct them.  

Table 2. Clinical response to ICI therapy in relation to defects in the antigen presentation machinery (case reports).

These tables are not properly inserted in the text. Authors need to correct them.  

Author Response

Dear reviewer,

Thank you very much for the positive and constructive feedback for the manuscript above.

The review article entitled "Response to immune checkpoint inhibitors is affected by deregulations in the antigen presentation machinery: a systematic review and meta-analysis" is very interesting. However, there are minor concerns regarding the Tables 1 and 2.  

Table 1. Clinical response to ICI therapy in relation to defects in the antigen presentation machinery (cohort studies).

Table 2. Clinical response to ICI therapy in relation to defects in the antigen presentation machinery (case reports).

Point 1: These tables are not properly inserted in the text. Authors need to correct them.  

Response 1: Thank you for pointing that out. We have improved the layout for better readability and hope for some additional editorial adjustments during the processing of the paper.

Point 2: Table 2. Clinical response to ICI therapy in relation to defects in the antigen presentation machinery (case reports).

Response 2: These tables are not properly inserted in the text. Authors need to correct them.  

We have improved this as well.

Kind regards
Maria Rasmussen,
Corresponding author.

Round 2

Reviewer 2 Report

My previous points have been addressed in this revised manuscript. 

Author Response

Thank you for your comments